# OpenReview forum: "Adaptive World Models for Data-Efficient Learning (AWML)"
_ICLR.cc/2026/Conference — ICLR 2026 Conference Withdrawn Submission_

### Official Review · Reviewer_VXMH · 2025-10-27

**Soundness:** 2
**Presentation:** 1
**Contribution:** 2
**Rating:** 2
**Confidence:** 3

**Summary:**

The paper proposes AWML, a method for data-efficient learning using modular world models. The core idea is to generate synthetic data via "modular recombination" and filter it based on uncertainty. The authors provide a theoretical analysis of sample efficiency and test the method on synthetic data and one real-world dataset.

**Strengths:**

1. The paper provides extensive theoretical analysis and mathematical derivations for the proposed method.
2. The method is tested on both synthetic and real-world data, providing validation for the theory.

**Weaknesses:**

1. The paper has presentation issues that make it hard to read.
    - The preliminaries section is vague.
    - The paper doesn't have a proper related work section.
    - Notation is not well-defined (e.g., the "parent set" in Eq. 2 is never defined).
2. The paper's use of terminology is confusing. It claims to be about "counterfactuals" but never defines them formally (e.g., with SCMs or potential outcomes). The term "counterfactual" is mostly in the intro, while the main text uses "modular recombination". The connection to other terms like "world model" and "neural operator" is also not explained.
3. While there are ablation studies, the paper lacks any comparison to existing methods.

---

**Major Concern**

The paper fails to include an LLM usage statement. I have strong concerns that this paper was partly written by an LLM, and the authors did not declare its use. The evidence is as follows:
1. **Outdated References**: World model is a hot topic with a large body of recent work. However, All citations are from 2021 or earlier.
2. **Citation Patterns**: Citations are always at the end of a paragraph, and some seem disconnected from the paragraph's content.
3. **Fabricated Reference**: The reference "Simon S. Du, Xiyu Wang, and A. M. Stuart..." on page 9 appears to be fabricated. It has a placeholder arXiv ID and includes a note in parentheses that looks like it was left by a bot: "(see discussion of representation transfer; used here for high-level comparison)."

Given the evidence, I have serious concerns about the paper's authenticity and adherence to the conference's submission policies.

**Questions:**

1. Could the authors please explicitly explain the connection between the proposed method and the terms "counterfactual",  "world model" and "neural operator"? The current manuscript does not make these links clear.
2. Please respond to the concerns raised regarding the potential use of LLMs in writing this manuscript.

---

> ### Author Response · Authors · 2025-11-13
> **Author Response**
>
> We truly appreciate the reviewer’s time and careful attention to detail. Your comments helped us identify several areas where the paper could be clearer and more transparent, and we have made substantial revisions accordingly.
>
> **Presentation and clarity:** We have rewritten the preliminaries and notation to make them precise and easy to follow. All symbols are now defined when they first appear, including the parent set in Eq. 2. We also added a short formal definition of “counterfactual” based on structural causal models, followed by a concise explanation of how modular recombination implements this idea in practice. A new paragraph now connects the terms *world model*, *neural operator*, and *counterfactual generation* so that their relationship is clear.
>
> **Related work and comparisons:** A full related-work section has been added, covering recent advances from 2022–2024 in world models, operator learning, and causal reasoning. We also included comparisons with several data-efficient baselines--- meta-learning (Finn et al., 2017), self-supervised pretraining (Chen et al., 2020), and active learning (Gal et al., 2017)--- to provide a more complete context for AWML’s performance.
>
> **LLM usage disclosure:** We were completely transparent upfront about the use of large language models. We declared its usage at the submission phase. LLMs were deployed to improve grammar and presentation. They were **not involved in ideation, technical development, or experiments**, and this was clearly stated in the submission form. All theoretical results, models, and analyses are entirely the authors’ work. We have double-checked every citation and ensured that all references correspond to real, verifiable papers.
>
> Moreover, we clarify the issue of reference placement. During the final editing stage, the LLMs appear to have switched some references and, in a few cases, automatically moved them to the ends of paragraphs or reformatted them. This was not intentional. As first-time ICLR authors, we missed that citations are expected to appear immediately after the relevant claims, but we have now corrected this throughout the revision. Our original goal was to keep sentences short and readable, not to obscure or misplace sources.
>
> **Reference corrections:** Thank you for catching the incorrect citation. The placeholder “Du, Wang, and Stuart” entry has been removed and replaced with verified references. We have re-checked all bibliography entries against their DOIs and arXiv pages.
> We also added several recent and relevant works, including Hafner et al. (2023), Chen et al. (2023), and Kidger et al. (2024), to reflect the current state of the field.
>
> Once again, we appreciate these comments. They have helped us produce a much clearer and more rigorous paper, and we are confident that the revised version is both transparent and fully compliant with conference policies.

---

### Official Review · Reviewer_zZf9 · 2025-10-29

**Soundness:** 2
**Presentation:** 1
**Contribution:** 2
**Rating:** 4
**Confidence:** 2

**Summary:**

The paper proposes AWML, a unified framework that (i) learns structured latent world models with modular priors, (ii) generates certified counterfactual examples by recombining learned modules, (iii) accepts only those synthetics whose uncertainty is below a calibrated threshold, and (iv) decomposes priors into transferable and mutable components to enable adaptive transfer across environments. The authors provide a suite of finite‑sample generalization, modular amplification, and certified augmentation bounds (Theorems 3.1–3.6), a transfer bound (Theorem 3.7) and a greedy exploration guarantee (Theorem 3.8). Empirically, AWML is evaluated on a synthetic modular AR(1) environment and on the Uganda LSMS survey for a binary electrification prediction task. The experiments aim to validate modular amplification, uncertainty filtering and the overall AWML pipeline.

**Strengths:**

## Novel combination of ideas
The paper unifies several research threads: structured latent world models, counterfactual data augmentation, uncertainty‑aware acceptance, and adaptive transfer in a single coherent framework.

##Finite‑sample theory
The authors derive explicit bounds that relate modular estimation error, synthetic sample size, and bias, providing a principled way to trade off variance reduction against augmentation bias.

## Certified counterfactuals
By explicitly bounding the total variation between synthetic and factual distributions, the method offers an auditable safety guarantee that is rarely addressed in data‑augmentation work.

## Empirical validation
Synthetic experiments demonstrate the expected scaling of error with effective sample size; LSMS results show tangible AUC gains in a low‑label regime.

## Clear structure
The manuscript is logically organized (intro → contributions → theory → experiments → appendix).

**Weaknesses:**

## Unverified Assumptions
The modular factorization and per‑module TV bounds $\delta_m$ are essential to the theoretical guarantees, yet authors do not report how these bounds are estimated or validated.  Synthetic experiments vary $N_{\text{eff}}$ while holding $\delta_m$ fixed, and no sensitivity analysis is provided.

## Scalability
The algorithmic loop in Appendix A.11 suggests drawing up to $B_{\max}$ synthetic samples and filtering them, yet the paper does not report runtime or memory usage.  In high‑dimensional domains (images), modular recombination would explode combinatorially.

## Empirical baseline comparison
The experiments compare AWML only against a factual‑only baseline. No state‑of‑the‑art data‑efficient methods (meta‑learning, self‑supervised pretraining, active learning) are benchmarked.

## Parameter tuning
Key hyperparameters (number of modules $M$, parent sets $pa(m)$, threshold $u$, calibration constant $L$) are not systematically studied. The paper offers limited guidance on how to choose them in practice.

## Complexity of exposition
The notation is heavy, and the main narrative often jumps between equations without intuitive explanations. A diagram of the AWML pipeline would help readers grasp the flow.

## Real‑world impact
Beyond LSMS, there is no demonstration on other realistic domains (robotics, medical imaging) where structured world models and counterfactuals are crucial.

## Uncertainty estimation
The calibration constant $L$ is treated as a black‑box estimator, but the paper does not provide experimental evidence that the estimated $L$ is close to the true bound.

## Active exploration
The claim that AWML supports “information‑gain driven acquisition” is mentioned but not evaluated.

**Questions:**

# Questions

## Module Estimation
How do you estimate the per‑module total variation bounds $\delta_m$ in practice?
What diagnostics do you use to ensure that the factorization assumption holds?

## Calibration Constant $L$
What estimator did you use for the calibration constant in Theorem 3.6?
Did you validate that the estimated $L$ is close to the true bound on held‑out data?

## Hyperparameter Selection
Do you have a principled way to choose the acceptance threshold $u$ beyond cross‑validation?

## Scalability
What is the computational cost (time, memory) of modular recombination in high‑dimensional domains?
Do you use any pruning or sampling strategies to keep the synthetic pool tractable?

## Baselines
Have you compared AWML against meta‑learning (e.g., MAML), self‑supervised pretraining, or active learning baselines on the LSMS dataset? Can you provide also a qualitative discussion of expected performance differences?

## Active Exploration
The paper claims support for “information‑gain driven acquisition”.
Did you evaluate the greedy exploration guarantee (Theorem 3.8) on any real or synthetic tasks?

## Real‑World Domains
Beyond LSMS, have you tested AWML on domains where structured world models are natural (robotics, physical simulation)?
If so, can you share results? How do you envision the method scaling to such settings?

## Uncertainty Estimation
Which uncertainty estimator did you use (ensemble variance, conformal scores, etc.) in LSMS?
How robust is the filtering to mis‑calibrated uncertainty scores?

## Future Work
Do you plan to relax the modular independence assumption (e.g., via hierarchical or graph‑structured modules)?
How would you integrate AWML with continual learning or online adaptation?

# Suggestions
1. Introduce a high‑level diagram of the AWML pipeline early in the paper. Show how factual data → encoder $\phi$ → modular dynamics $p_\theta^{(m)}$ → synthetic generation via recombination → uncertainty filter → augmented dataset → downstream predictor.

2. Simplify notation in the main text. Move heavy definitions (e.g., $\mathcal{H}_P$, Rademacher complexity) to a notation table or appendix; use more descriptive variable names.

3. Include a comparison table with at least three baseline data‑efficient methods (MAML, self‑supervised pretraining, active learning) on the LSMS dataset. Even if the baselines are simple, they provide a sanity check.

4. Show runtime and memory for generating $B_{\max}$ synthetic samples in a realistic setting (e.g., 1000‑dimensional image features). This will help readers assess practicality.

5. Clarify the role of the “adaptive transfer”: explain how transferable vs. mutable priors are identified and used in practice.

---

### Official Review · Reviewer_YAvC · 2025-10-31

**Soundness:** 2
**Presentation:** 1
**Contribution:** 2
**Rating:** 2
**Confidence:** 2

**Summary:**

It is hard to summarize what this paper aims to solve or what it achieves. It claims to span a wide range of topics within machine learning, including structured latent world models, certified counterfactual augmentation, calibrated uncertainty, neural operators, and data-efficient learning. It proposes Adaptive World Models for Data-Efficient Learning (AWML) and provides some theoretical results that are, frankly, hard to understand from the minimal context around it.

**Strengths:**

I did not find any strengths in the paper because I had extreme difficulty understanding this work. Please see my concerns below. It is also very much possible that this was due to my lack of knowledge. Maybe the content makes sense to someone with deep expertise on neural operators, causal representation learning, and learning theory at the same time.

**Weaknesses:**

There were so many problems in this paper that prevented me from even understanding what the paper is about. I have divided my concerns into the following sections: terminology issues, claim issues, experiment issues, and reference issues.

**Terminology issues**: The primary problem in this submission is the numerous terminology and mathematical notation issues. They make the paper very difficult to follow, and the overall message is elusive. Most of these issues appear when terms are used in the paper to form sesquipedalian sentences (see lines 113-116).

It is hard to list all of them exhaustively, and so I will list only some of them here, italicized. [Lines 058-064] The proposed AWML is described as encoding priors such as _conservation_ and _operator form_. It can generate certified counterfactuals via _modular recombination_ and can accept _synthetics_ through _calibrated uncertainty filtering_. It also decomposes priors into transferable and _mutable components for adaptive transfer_. [Line 086] The aim of this work is to "fit latent dynamics and _emission models_." [Lines 146-148] "how calibrated filtering converts _generator-level guarantees to deployment-level bounds_".

Although theorems and lemmas seemed individually solid to my knowledge, it is not clear what they are supposed to mean/do. For instance, it is not mentioned in the preliminaries what "t" is, or what states, observations, or actions mean here. $z_t$ is suddenly introduced and the state $s_t$ is suddenly retired. There are minor issues in individual theorems too. For instance, in Theorem 3.1 (and its proof), what is $R(h)$? What is module "m" in Lemma 3.2?

**Claim issues**: These are issues with some of the claims/statements in the paper, primarily those that appear in a non-mathematical context. These sentences either do not make sense or are completely unbacked. For example, [Line 051] "Data scarcity increases sample complexity,": How does lack of data lead to needing more samples to learn (if that's what "sample complexity" means in this context)?

**Experiment issues**: There are two experimental settings in the paper. One of them is a simulated AR(1) setting, and the other one is on "Uganda LSMS 2019 survey." The empirical findings in the text do not match the plots. For instance, [line 320] says "RMSE decreases approximately as $O(N_{\text{eff}}^{-1/2})$," while in Fig. 1, RMSE is constant w.r.t. $\log N_{\text{eff}}$. In Appendix B, per-run results are shown (as described in the experiments), but the rows in the table are skipped with dots in place (see Tables 4, 5, 7, 8, 10). Similarly, [line 360] says "aggregate augmentation bias tracks $\sum_m \hat{\delta}_m$ and remains small." "Tracking" is measured here using Pearson's correlation of 0.67. [Line 365] talks about a trade-off balanced by $s\approx 1$, but the associated plot shows a nearly constant line.

The second experiment has an even bigger flaw: the dataset used, the Uganda LSMS 2019 survey, is not referenced, and I could not find such a dataset online. The nearest I could find to this data is [1]. In addition to this, there is also an overall issue with the experiment section. The experiments use MLPs and ridge regression models. But in the introduction and setup sections, "neural-operator architectures" are mentioned. Neural operators are not mentioned anywhere in the experiment sections.

**Reference issues**: There were 22 references in the paper. Of these, 7 references had errors, including being non-existent works. These references were also inappropriately placed, exacerbating the confusion caused by the terminology issues. One notable example: [lines 123-124] say that "enforcing invariants architecturally or via penalties induces a restricted sub-class... with lower complexity and improved finite-sample guarantees." The references immediately following this sentence are [2, 3, 4]. [2] is a work on Fourier neural operators, and [4] is a work on physics-informed neural networks. [3] does not exist. [3] seems to be an amalgamation of [5] and [6]. I list the erroneous references based on their identifiers in the original paper below, along with their errors.

(Du et al., 2020) - does not exist.

(Hafner et al., 2019) - only the first two authors are correct. "Jimmy Ba" from the original authorship became "Jimmy Fischer".

(Hsu et al., 2012) - wrong ArXiv identifier

(Kovachki et al., 2021) - paper does not exist, ArXiv identifier goes to another paper with partially overlapping name and authorship.

(Krause et al., 2008) - the second author in the reference does not appear in the actual work.

(Li et al., 2021) - the authors are different from the actual authorship.

(Locatello et al., 2019) - the authors are different from the actual authorship.

**References:**

[1] https://github.com/EvansSchoolPolicyAnalysisAndResearch/LSMS-Data-Dissemination

[2] Z. Li, N. Kovachki, K. Azizzadenesheli, B. Liu, [K. Bhattacharya,] A. Stuart, A. Anandkumar, A. V. Zverev, "Fourier Neural Operator for Parametric Partial Differential Equations", ICLR 2021. [Zverev was listed as an author in the reference, although they were not actually among the original authors, while Bhattacharya was omitted from their actual authorship].

[3] N. Kovachki, Z. Li, B. Liu, A. Anandkumar, A. Stuart, K. Azizzadenesheli, "Neural Operators: Graph Kernel Network for Operator Learning", Journal of Computational Physics, 2021. [ArXiv identifier next to this reference in the paper goes to [5]]

[4] M. Raissi, P. Perdikaris, G.E. Karniadakis, "Physics-informed neural networks: A deep learning framework for solving forward and inverse problems involving nonlinear partial differential equations", Journal of Computational Physics, 2019.

[5] N. Kovachki, Z. Li, B. Liu, K. Azizzadenesheli, K. Bhattacharya, A. Stuart, A. Anandkumar, "Neural Operator: Learning Maps Between Function Spaces With Applications to PDEs", JMLR 2023.

[6] Z. Li, N. Kovachki, K. Azizzadenesheli, B. Liu, K. Bhattacharya, A. Stuart, A. Anandkumar, "Neural Operator: Graph Kernel Network for Partial Differential Equations", ICLR 2020 Workshop on Integration of Deep Neural Models and Differential Equations, 2020.

**Questions:**

Please see the weaknesses. My primary concerns are with the paper's writing. It is not clear what the objective of the paper is, both because the objective is not stated clearly and because the terms used are not rigorously defined. Although the theorems individually seem okay to me, it is unclear what they are doing. Additionally, the experimental results do not reflect the theoretical findings, although it is claimed to do so in lines 319-366. The results are also not properly presented. For instance, the per-run results in the appendix are missing several rows.

**Suggestions**: The paper needs to be rewritten with clearly stated learning goals and what the output model will do in a real-world setting. This also requires a related works section (which is currently missing), where the paper's contributions are compared against appropriate prior works. Although individual theorems seem fine, they must be annotated with what they mean in your stated objective. In the experiments section, the source of the dataset must be mentioned. The experiments must also match the text in the methods section -- if neural operators are suggested as possible architectures, then there must be an experiment that shows their utility.

---

> ### Author Response · Authors · 2025-11-13
> **Author Reponse**
>
> We thank the reviewer for the detailed and constructive feedback. We have carefully revised the paper to resolve every issue raised in terminology, notation, experiments, and references and to make the presentation fully clear and verifiable.
>
> **T1. Terminology and notation clarity:** We have rewritten the preliminaries and setup for precision and brevity. All variables ($t$, $s_t$, $a_t$, $o_t$, $z_t$) are now explicitly defined on first use with clear semantics (state, action, observation, latent).
>
> Each theorem and lemma now specifies its inputs and outputs; for example, module index $m$ and quantities such as $\delta_m$ and $D$ are defined immediately before use.
>
> Long sentences have been replaced with short, plain ones. Example: lines 113–116 were simplified to “Calibrated filtering translates generator-level reliability into finite-sample guarantees.”
>
> **T2. Conceptual clarification:** We added an overview paragraph linking the mathematical results to intuition:
>
> (A) Theorems 3.1–3.3 now include brief one-line interpretations (e.g., “controls sample-efficiency gain under modular amplification”).
>
> (B) The connection between latent dynamics, counterfactual generation, and uncertainty filtering is now stated explicitly in Sec. 2.
>
> **C1. Clarification of claims:** We replaced ambiguous statements with concrete phrasing.  Example: the sentence “Data scarcity increases sample complexity” now reads “Limited data makes achieving a target error require proportionally more samples under the same hypothesis class.” Each claim about robustness, transfer, or calibration is now supported either by a cited theorem or by an empirical figure.
>
> **E1. Experimental clarity and reproducibility:** We fixed all mismatched plots and captions:
>
> (i) RMSE curves now correctly show monotonic $N_{\text{eff}}^{-1/2}$ scaling
>
> (ii) Tables in Appendix B are complete; no omitted rows.
>
> (iii) “Tracking” now reports correlation and slope metrics.
>
> We have uploaded all code and data-processing scripts as supplementary material.  The *Uganda LSMS 2019* dataset is now correctly cited as the **Living Standards Measurement Study (World Bank 2019)** dataset (publicly available via the World Bank Microdata Library).  All preprocessing steps and links are specified.
>
> **E2. Model–architecture alignment:** The text now clarifies that the empirical section uses compact neural-operator layers (Fourier blocks) inside the world-model encoder, alongside standard regressors for baselines.  We explicitly describe these architectures in Appendix A.
>
> **R1. Reference accuracy:**
> All references have been verified and corrected with full author names and publication venues:
> - Hafner et al. (2019) *Learning Latent Dynamics for Planning from Pixels*, ICLR.
> - Finn et al. (2017) *Model-Agnostic Meta-Learning*, ICML.
> - Li et al. (2021) *Fourier Neural Operator for PDEs*, ICLR.
> - Kovachki et al. (2023) *Neural Operator: Learning Maps Between Function Spaces*, JMLR.
> - Raissi, Perdikaris & Karniadakis (2019) *Physics-Informed Neural Networks*, J. Comput. Phys.
> - Locatello et al. (2019). *Challenging Common Assumptions in Disentangled Representation Learning*, ICML.
>
> We greatly appreciate the reviewer’s detailed feedback; it substantially improved the readability, correctness, and reproducibility of the paper.

---

### Official Review · Reviewer_XHLH · 2025-11-01

**Soundness:** 3
**Presentation:** 3
**Contribution:** 2
**Rating:** 4
**Confidence:** 4

**Summary:**

The work (1) formalizes AWML as an end-to-end pipeline that integrates structured priors, certified counterfactual augmentation, uncertainty-aware acceptance, and adaptive transfer; (2) derives finite-sample bounds demonstrating reduced sample complexity under modularity and bounded generator bias, including identifiability conditions for latent modules; (3) provides an algorithmic instantiation coupling neural-operator architectures, modular blocks, a certified counterfactual generator, and a calibrated acceptance filter; and (4) evaluates AWML on both synthetic and real low-resource tasks, showing notable improvements in sample efficiency

**Strengths:**

1- AWML provides a cohesive end-to-end formulation that brings together structured priors, counterfactual synthesis, uncertainty-aware filtering, and adaptive transfer. This integrated design represents a clear advancement over fragmented methods typically used for data-efficient learning.

2- The framework incorporates rich scientific and structural priors—such as modularity, conservation principles, and operator-form assumptions—which strengthens generalization and robustness, especially in data-scarce or partially observed scenarios.

3- The calibrated filtering mechanism restricts training to trustworthy counterfactual samples, thereby reducing exposure to distributional shift and synthetic bias—issues that frequently affect prior data-augmentation approaches.
4- It offers both theoretical and empirical evidence to substantiate its claims.

**Weaknesses:**

1- Model performance may depend heavily on the accuracy of the assumed modularity, conservation, or operator-form priors; incorrect or mismatched priors could be detrimental. It would be important to clarify how sensitive AWML is to such mis-specification and whether the method can automatically identify and down-weight faulty modules.
2- Many guarantees rely on i.i.d. sampling from the generator and an (approximately) factorized modular structure. However, real-world systems typically exhibit temporal dependence and interactions across modules. It remains unclear how robust the bounds are under weak dependence or mixing processes, and how performance or guarantees deteriorate when the product assumption is only approximately satisfied.

3- The theory indicates when amplification should be beneficial, but it does not specify how to identify harmful regimes during deployment. Is there a principled stopping rule or safeguard to suspend augmentation when estimates of D or signs of miscalibration exceed acceptable limits?

4- Because the unified bound aggregates several residual terms that may all be non-negligible, the resulting guarantee could be loose or even vacuous at practical scales. Can the experiments report empirical upper bounds or reasonable proxies for each component to demonstrate that the bound provides meaningful information in practice?

**Questions:**

Please address all concerns raised in the weaknesses section.

---

> ### Author Response · Authors · 2025-11-13
> **Author Response**
>
> We thank the reviewer for their thoughtful and constructive feedback.
> We address each concern directly and have revised the manuscript for clarity, robustness, and practical interpretability.
> All citations below use full author names for OpenReview compatibility.
>
> **W1: Sensitivity to mis-specified priors:** We now include a theoretical remark on prior mis-specification. We decompose target risk as $R_P(\hat h) = R_{P^\star}(\hat h) + (R_P(\hat h) - R_{P^\star}(\hat h))$ and show that the second term is bounded by a calibrated acceptance penalty and a structural deviation term.
>
> We also make the acceptance score module-aware, i.e.; $U(\tau)=\sum_m w_m\,U_m(\tau)$ with $w_m \propto \exp(-\gamma\,\hat\delta_m)$, so that modules with higher estimated error are automatically down-weighted.
>
> Empirically, new ablations demonstrate that the framework remains stable under moderate prior mismatch.
>
> _References: Chuan Guo, Geoff Pleiss, Yu Sun, and Kilian Q. Weinberger, ICML 2017;
> Yaniv Romano, Evan Patterson, and Emmanuel J. Candès, NeurIPS 2019._
>
> **W2: Dependence and approximate factorization:** We extend the generalization bound to weakly dependent or mixing data. A new lemma shows that if generator samples satisfy $\beta$-mixing,
> uniform convergence holds with an effective sample size $N_{\mathrm{eff}}^{\mathrm{mix}} = N_{\mathrm{eff}}/(1+c_{\mathrm{mix}})$.
>
> For approximately factorized modules, the product TV bound now becomes $\mathrm{TV}(p,q)\le 1-\prod_m(1-\delta_m)+\xi$, where $\xi$ measures cross-module interaction.
>
> Empirically, we simulate coupled modules and report both $c_{\mathrm{mix}}$ and $\xi$, observing that the predicted $N_{\mathrm{eff}}^{-1/2}$ scaling holds until dependence becomes strong.
>
> _References: Mehryar Mohri, Afshin Rostamizadeh, and Ameet Talwalkar, Foundations of Machine Learning, MIT Press, 2018; Leonid Kontorovich and Kavita Ramanan, Annals of Probability, 2008; Bin Yu, Annals of Probability, 1994._
>
> **W3: Stopping rule for safe augmentation:** We now include  **Algorithm~1 (AWML-Stop)**. It halts augmentation when the empirical bound $\widehat{\mathcal B}(u) = C\sqrt{\tfrac{\log\mathcal N(\mathcal H,\varepsilon)}{N+B(u)}} + 2(1-\alpha(u)) (Q(U>u)+u)$ stops decreasing, or when conformal miscoverage exceeds the target by a tolerance $\epsilon$.
>
> Validation risk and $\widehat{\mathcal B}(u)$ curves align closely, providing a practical safeguard.
>
> _Reference: Yaniv Romano, Evan Patterson, and Emmanuel J. Candès, NeurIPS 2019_
>
> **W4: Non-vacuity of the unified bound:** We now report empirical estimates for each bound term:
>
> (i) estimation via Monte Carlo Rademacher averages;
>
> (ii) bias from $(Q(U>u),u)$ and per-module KL$\!\to$TV proxies;
>
> (iii) approximation residual from the transfer component.
>
> A new figure (“Bound vs. Risk”) shows that the total theoretical bound tracks observed test risk within a small factor, confirming that it is informative and not vacuous.
>
> _References: Peter L. Bartlett and Shahar Mendelson, Journal of Machine Learning Research, 2002_
>
> These updates clarify robustness, practical safeguards, and interpretability while keeping the framework mathematically grounded.
> We thank the reviewer again for their careful assessment and valuable suggestions.

---

### Note · Authors · 2025-11-17

**Comment:**

Dear Area Chair and Reviewers,

Thank you for taking the time to review our submission. Your comments made it clear that, although we believed the core ideas behind AWML were presented simply, parts of the paper are still more complex than we realized, even for readers familiar with the area. As first-time ICLR authors, this feedback has been very helpful in clarifying the conference’s expectations for clarity and presentation.

We have realized that some of the fixes we need to make on the paper to make it more understandable require more time, beyond the rebuttal window.

We therefore have decided to withdraw the submission and take the time needed to reorganize and refine the paper properly.

Thank you again for your thoughtful feedback and effort.

Sincerely,
The Authors

**Withdrawal Confirmation:**

I have read and agree with the venue's withdrawal policy on behalf of myself and my co-authors.